

# MemCat: a new category-based image set quantified on memorability

Lore Goetschalckx and Johan Wagemans

Brain & Cognition, KU Leuven, Leuven, Belgium

## ABSTRACT

Images differ in their memorability in consistent ways across observers. What makes an image memorable is not fully understood to date. Most of the current insight is in terms of high-level semantic aspects, related to the content. However, research still shows consistent differences within semantic categories, suggesting a role for factors at other levels of processing in the visual hierarchy. To aid investigations into this role as well as contributions to the understanding of image memorability more generally, we present MemCat. MemCat is a category-based image set, consisting of 10K images representing five broader, memorability-relevant categories (animal, food, landscape, sports, and vehicle) and further divided into subcategories (e.g., bear). They were sampled from existing source image sets that offer bounding box annotations or more detailed segmentation masks. We collected memorability scores for all 10 K images, each score based on the responses of on average 99 participants in a repeat-detection memory task. Replicating previous research, the collected memorability scores show high levels of consistency across observers. Currently, MemCat is the second largest memorability image set and the largest offering a category-based structure. MemCat can be used to study the factors underlying the variability in image memorability, including the variability within semantic categories. In addition, it offers a new benchmark dataset for the automatic prediction of memorability scores (e.g., with convolutional neural networks). Finally, MemCat allows the study of neural and behavioral correlates of memorability while controlling for semantic category.

## INTRODUCTION

A large body of work within the visual memory field has been devoted to questions about its capacity and fidelity (for a review, see *Brady, Konkle & Alvarez, 2011*). Often, these studies abstract away from the properties of the to-be-remembered stimuli and potential differences between them. Yet, work by *Isola et al. (2014)*, using everyday images, showed that they do not all share the same baseline likelihood of being remembered and recognized later. Instead, images differ in "memorability" in ways that are consistent across participants and this can be measured reliably (*Isola et al., 2014*).

To assess memorability, *Isola et al. (2014)* used a repeat-detection memory task, in which participants watch a sequence of images and respond whenever they see a repeat of a previously shown image. The researchers assigned a memorability score to 2,222 scene

Corresponding author
Lore Goetschalckx,
lore.goetschalckx@kuleuven.be

images based on the proportion of participants recognizing the image upon its repeat. They found that memorability rank scores were highly consistent across participants. In other words, there was a lot of agreement as to which images were remembered and recognized, and which ones were easily forgotten. This suggests that memorability can indeed be considered an intrinsic image property and that whether you will remember a certain image does not only depend on you as an individual, but also on the image itself. The result has furthermore been replicated with a more traditional long-term visual memory task, with a separate study and test phase (*Goetschalckx, Moors & Wagemans, 2018*). Moreover, image memorability rankings have been shown to be stable across time (*Goetschalckx, Moors & Wagemans, 2018*; *Isola et al., 2014*), across contexts (*Bylinskii et al., 2015*), and across encoding types (intentional versus incidental; *Goetschalckx, Moors & Wagemans, 2019*). Finally, while they might be related to some extent, image memorability does not simply boil down to popularity (*Khosla et al., 2015*), aesthetics (*Isola et al., 2014*; *Khosla et al., 2015*), interestingness (*Gygli et al., 2013*; *Isola et al., 2014*), or the ability of an image to capture attention (*Bainbridge, 2017*).

The findings spurred new research aimed at understanding and predicting memorability. When it comes to merely predicting the memorability score of an image, the best results so far are achieved using convolutional neural networks (CNNs; e.g., *Khosla et al., 2015*). When it comes to truly understanding, on the other hand, CNNs have often been critiqued to be black boxes (however, see *Benitez, Castro & Requena, 1997* for counterarguments). It is not always clear to us humans why a CNN predicts a certain score for one image and not another. Nonetheless, *Khosla et al.*'s (*2015*) analyses provided some further insight, mostly pointing at differences between broader image categories and content types. For example, units in the network displaying the highest correlation with memorability seemed to respond mostly to humans, faces, and objects, while those with the lowest correlation seemed to respond to landscapes and open scenes. Furthermore, the most memorable regions of an image, according to the CNN, often capture people, animals or text. These findings are in line with earlier work, which also predominantly revealed high-level semantic attributes. *Isola et al. (2014)*, for example, showed that the predictive performance of a model trained on mere object statistics (e.g., number of objects) was boosted considerably when the object labels were taken into account. In addition, a model trained on the overall scene labels alone, already predicted memorability scores with a Spearman's rank correlation of .37 to the ground truth. Memorable images often had labels referring to people, interiors, foregrounds, and human-scaled objects, while labels referring to exteriors, wide-angle vistas, backgrounds, and natural scenes were associated with low image memorability scores.

Together, these findings suggest that a fair share of the variability in memorability resides in differences between semantic categories. Perhaps this is not surprising considering the central position occupied by categories in the broader cognitive system. It has been said that carving up the world around us into meaningful categories of stimuli that can be considered equivalent is a core function of all organisms (*Rosch et al., 1976*). It helps us understand novel events and make predictions (*Medin & Coley, 1998*). According to *Rosch et al. (1976)*, categories are represented hierarchically and are organized into a taxonomy

of different levels of abstraction. The basic level is the best compromise between providing enough information and being cognitively inexpensive. It is also the preferred naming level (e.g., "cat"). Other levels can be superordinate (e.g., "feline" or "mammal") or subordinate (e.g., "tabby"). Recently, *Akagunduz, Bors & Evans (2019)* pointed out that categories are also used to organize memory. More specifically, instead of encoding an image as a mere collection of pixels, we extract visual memory schemas associated with its category (i.e., key regions and objects and their interrelations), along with an image's idiosyncrasies. To map these visual memory schemas, they had participants indicate which image regions helped them recognize the image. The resulting maps showed high consistency across participants, suggesting that visual memory schemas partly determine what participants find memorable. Moreover, humans can visually categorize an object depicted in an image very rapidly and accurately (e.g., *Bacon-Macé et al., 2005*; *Fei-Fei et al., 2007*; *Greene & Oliva, 2009*), as well as categorize the image at the level of the whole scene (e.g., *Delorme, Richard & Fabre-Thorpe, 2000*; *VanRullen & Thorpe, 2001*; *Xu, Kankanhalli & Zhao, 2019*). Often a single glance suffices. Interestingly, *Broers, Potter & Nieuwenstein (2017)* observed enhanced recognition performance for memorable versus non-memorable images in an ultra-rapid serial visual presentation task. Finally, there is also evidence for a category hierarchy in the representations in high-level human visual cortex, with for example clusters for animacy and subclusters for faces and body parts (*Carlson et al., 2013*; *Cichy, Pantazis & Oliva, 2014*; *Kriegeskorte et al., 2008*).

While semantic categories (or labels) seem to play a large role in image memorability, they do not explain all the observed variability. Interestingly, consistent differences in memorability scores remain even *within* image categories (*Bylinskii et al., 2015*). *Goetschalckx et al. (2019)* for example, have argued that part of that variability might be due to differences in how well the image is organized. Nonetheless, the concept of memorability and its correlates are not yet fully understood to date and further research is required to paint a clearer picture. The current work presents a novel, category-based dataset of images quantified on memorability, designed for research to achieve this goal (example images in Fig. 1).

To the best of our knowledge, there were previously five large image sets with memorability scores, three consisting of regular photographs, which is also the focus here: *Isola et al. (2014)*, FIGRIM (*Bylinskii et al., 2015*), and LaMem (*Khosla et al., 2015*), and two more specialized sets, which we will not further discuss here: *Bainbridge, Isola & Oliva* (*2013*; face images), and *Borkin et al.* (*2013*; data visualizations). For completeness, we also mention a smaller set (850 images) that was used to study which objects in an image are memorable (*Dubey et al., 2015*). Table 1 compares MemCat to the other large datasets. The comparison is discussed in more detail below.

A first feature of the current dataset is its hierarchical category structure. It was designed to be representative for five different broad natural categories and to allow the study of memorability differences within semantic categories. The set is characterized by a hierarchy of five broader categories, further divided into more fine-grained subcategories. Only the FIGRIM set also offers a category structure, but the number of exemplar images per category was lower: 59–157, compared to 2,000 in the current set. We opted for broad categories to
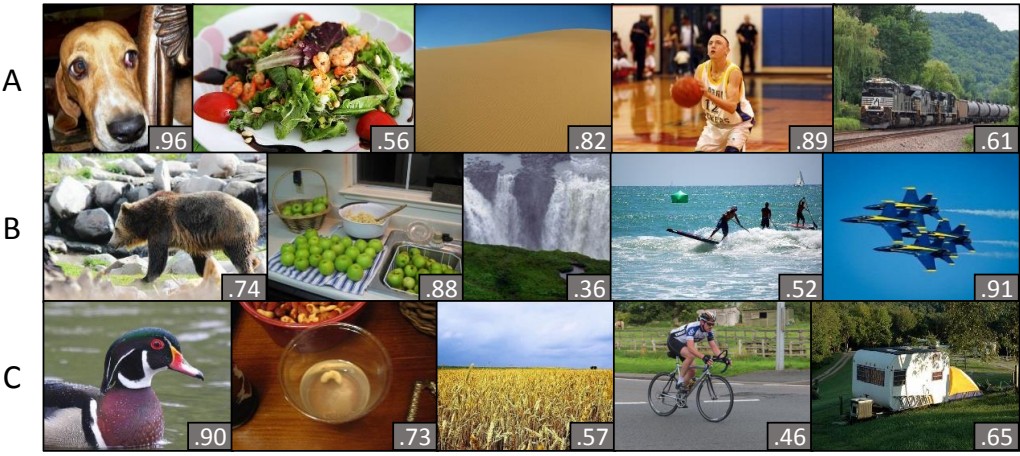

**Figure 1** **Example images of MemCat.** The memorability score, calculated as the hit rate across participants (corrected for false alarms; $(H-F)/N_{resp}$) is indicated in the bottom right corner. In line with previous research, images differed consistently in their memorability score, even within semantic categories. MemCat represents five broader semantic categories: animal, food, landscape, sports and vehicle. Each row (A–C) displays exemplar images in that category order.

**Table 1** **Comparison MemCat to other memorability datasets.**

|  | *Isola et al. (2014)* | FIGRIM | LaMem | MemCat |
|---|---|---|---|---|
| Category-based | No | Yes | No | Yes |
| Number of quantified images | 2,222 | 1,754 | ∼60 K | 10 K |
| Bounding boxes or segmentation data | Yes | Yes | No | Yes |

ensure that the whole was still varied and representative enough, while containing a large number of exemplar images per category at the same time. Moreover, our final choice of categories: animal, food, landscape, sports, and vehicle, was motivated by their relevance for memorability, meaning that they (or related categories) have been observed to differ in their overall memorability in previous research (*Isola et al., 2011*; *Isola et al., 2014*; *Khosla et al., 2015*). For example, knowing that the presence of people in an image is predictive for memorability (see above), we chose one category of images depicting people as the main subject and avoided including images with people in other categories. For this one category, we chose "sports", because "people" in itself constitutes a category that was too broad in comparison to the other categories and did not lend itself well for a division into subcategories. Furthermore, we included an animal category as a non-human animate category, food and vehicle as more object-based categories, and landscape to represent the wide exteriors that are often associated with lower memorability scores (*Isola et al., 2011*; *Isola et al., 2014*; *Khosla et al., 2015*).

Second, we aimed for a large set, such that it would be suitable for machine learning approaches. With a total of 10,000 images quantified on memorability, the current set is the second largest memorability dataset, after LaMem.
Third, we sampled images from existing datasets, such that the image annotations collected there would also be available for researchers studying memorability. In particular, we searched for images annotated with segmentation masks or at least bounding boxes, reasoning that they may hold some indications of how the image is organized (e.g., where is the subject located), which might be of particular interest when studying memorability within categories and factors other than semantics.

In summary, the unique combination of features of MemCat, together with its richness in data, make it a valuable addition to the literature. Among the possible uses by memorability researchers are the study of what makes an image memorable beyond its category, a benchmark for machine learning approaches, and a semantically controlled stimulus set for psychophysical or neuroscientific studies about the correlates of memorability (elaborated in the 'Discussion' section). However, given that categorization is a core function of the human mind, MemCat would also appeal to a much broader range of cognitive (neuro)scientists.

## MATERIALS & METHODS

### Participants

There were 249 undergraduate psychology students (KU Leuven) who participated in this study in exchange for course credits (216 female, 32 male, 1 other). Four students did not disclose their age and the remainder were aged between 18 and 27 years old ($M = 19.24$, $SD = 0.94$). The majority of the participants, however, were recruited through Amazon's Mechanical Turk (AMT) and received a monetary compensation (see further for details). The settings on AMT were chosen such that only workers who indicated to be at least 18 years old and living in the USA could participate. Further eligibility criteria were that the worker had to have an approval rate of at least 95% on previous human intelligence tasks (HITs) and a total number of previously approved HITs of at least 100. A total of 2162 AMT-workers participated in this study (1,139 female, 917 male, 4 other, and 102 who did not disclose this information). For the 1851 workers who disclosed their age, the reported ages ranged between 18 and 82 years old ($M = 37.14$, $SD = 11.89$). The AMT data collection took place from April 2018 till July 2018. Data collected through AMT has been shown to come from participant samples that are more diverse than student samples and to be comparable in quality and reliability to those collected in the lab (e.g., *Buhrmester, Kwang & Gosling, 2011*).

### Materials

MemCat consists of 10,000 images sampled from four previously existing image sets: ImageNet (*Deng et al., 2009*), COCO (*Lin et al., 2014*), SUN (*Xiao et al., 2010*), and The Open Images Dataset V4 (*Kuznetsova et al., 2018*). The four source sets were chosen because of their large size (i.e., number of images), the availability of semantic annotations, and the availability of bounding box annotations or more complete segmentation masks for at least a subset of their images. The images selected from the source sets to be included in MemCat belonged to the five broader semantic categories outlined in the Introduction: animal (2,000 images), food (2,000 images), landscape (2,000 images), sports (2,000 images), and vehicle

(2,000 images). We explain the different steps in the selection procedure in more detail below.

As a first step, we listed at least 20 subcategories for each broader category. The goal was to obtain 2,000 images per category, without including more than 100 exemplar images per subcategory. This was to ensure a reasonable level of variability and to avoid high levels of false alarms in the memory task (see further). The subcategories were then translated to semantic annotations from the source dataset. For example, for the subcategory "bear" (animal), we used COCO images annotated with a "bear" tag and ImageNet images from nodes "American black bear". "brown bear", and "grizzly". An overview of our hierarchy of categories and subcategories, can be found in Fig. 2.

The second step consisted of automatically sampling exemplar images from the listed subcategories, while satisfying a number of shape restrictions. To avoid that images would stand out because of an extreme aspect ratio, we only sampled images with aspect ratios between 1:2 and 2:1. Furthermore, the minimum resolution was set to 62,500 pixels. Finally, to ensure that the images would fit comfortably on most computer monitors, we adopted a maximum height of 500 pixels and a maximum width of 800 pixels. However, for SUN and The Open Images Dataset, only a low number of images satisfied the latter two restrictions (they were often too big), which is why we opted to resize (using Hamming interpolation) images from those two source datasets to meet the restrictions. Apart from the shape restriction, we also restricted the sampling to images for which bounding box annotations or more complete segmentation masks were available from the source datasets. Finally, we sampled more images than the target number (2,000 images per broad category), anticipating exclusions in the next step.

The third step constituted manual selection work, carried out by the first author, assisted by two student-interns. We manually went through the exemplar images sampled in the previous step, and eliminated images following a number of exclusion rules. The exclusion rules can roughly be divided into two kinds. A first kind of exclusion rule touches upon the quality of the image. We excluded images of poor image quality (e.g., very dark, very much overexposed, blurry, etc.), images that did not convincingly belong to the subcategory they were assigned to,[1] images in greyscale or looking like they were the result of another color filter, images that were not real photographs (e.g., drawings, digitally manipulated images, computer generated images), and collages. A second set of rules concerns factors that could affect the memorability of an image, but were not of interest for the purpose of MemCat. One such factor is text. We excluded images containing large, readable text or text not belonging to the image itself (e.g., date of capture). Another factor was the presence of people in the image. There was one designated "people" category, the sports category, meaning that every included exemplar image depicted one or more people practicing sports. However, the presence of people was avoided in all other categories (but we allowed anonymous people in the background in the vehicle category or the presence of a hand in images of the food category). Furthermore, images depicting remarkably odd scenes (e.g., dog wearing Santa Clause costume) were also excluded. Similarly, we avoided images depicting famous places or people (e.g., Roger Federer or Cristiano Ronaldo in the sports category), and images of dead, wounded or fighting animals. In addition to these exclusion

[1]This could happen, for example, with images from the COCO source dataset. COCO images do not come with a single, overall scene label, but instead come with multiple semantic tags describing what is in the image. For this reason, an image annotated with the tag "cat", for instance, could be more of a living room image that just happens to have a cat sleeping somewhere in a corner in the background.

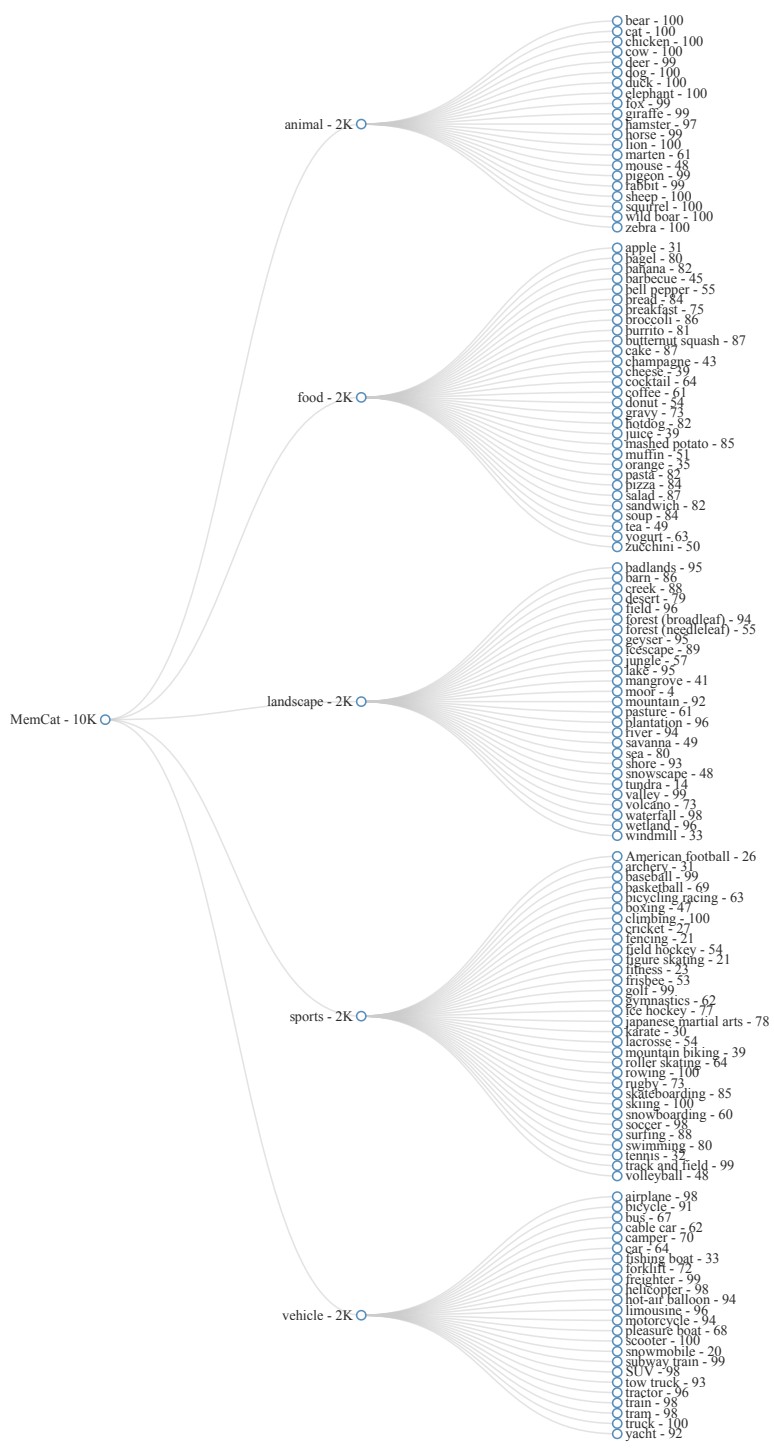

**Figure 2** Category hierarchy of MemCat.

rules, we also tried, to the best of our ability, not to include (near) duplicate images. If the target number of images was not obtained after Step 3, we reverted back to Step 2, if there were still images to sample from, or to Step 1 if we needed to include additional subcategories.

Finally, for those categories for which more than the target number of images survived Step 3, there was a fourth step to randomly down-sample the selection to the target number, assigning higher sampling probabilities to images annotated with segmentation masks.

In addition to MemCat, we collected 10,000 filler images, that were not quantified on memorability themselves, but were needed in the memory task used to quantify the other images. The filler images were sampled randomly from The Open Images Dataset, but from a different subset to avoid overlap.[2] As these images would function only as filler images, there were fewer restrictions. For example, the images could be of any category, they were allowed to contain text, etc. However, the same shape restrictions were still applied.

## Procedure

Having carefully collected 10,000 images for MemCat, the next step was to quantify them on memorability. Following previous work, this was achieved by presenting the images in an online repeat-detection memory game (*Isola et al., 2014*; *Khosla et al., 2015*), in which participants watch a sequence of images and are asked to respond when they recognize a repeat of a previously shown image. Students participating for course credits played the game in the university's computer labs, hosting about 20 students at a time. AMT workers played the game from the comfort of their homes (or whichever location they preferred). Prior to starting the game, all participants were prompted to read through an informed consent page explaining the aims of the study and their rights as participants. They could give their consent by actively ticking a box. The study was approved by SMEC, the Ethical Committee of the Division of Humanities and Social Sciences, KU Leuven, Belgium (approval number: G-2015 08 298).

For the task design of the game, we closely followed *Khosla et al. (2015)*, as their version of the game was designed to quantify large numbers of images. We divided the game into blocks of 200 trials. On each trial, an image was presented at the center of the browser window for a duration of 600 ms, with an intertrial interval of 800 ms. During this interval, a fixation cross was shown. Sixty-six images were target images, sampled randomly from MemCat, and repeated after 19 to 149 intervening images. Forty-four images were random filler images that were never repeated. Finally, there were 12 additional random filler images that were repeated after 0 to 6 intervening images to keep participants attentive and motivated. They are referred to as vigilance trials. Participants could indicate that they recognized a repeat by pressing the space-bar. They did not receive trial-by-trial feedback, but were shown their hit rate as well as number of false alarms at the end of the block. Figure 3 presents a schematic of the game.

Each block lasted a little less than 5 min. Care was taken to ensure that an image was never repeated more than once and never across blocks. Students were asked to complete as many blocks as they could in one hour, with one bigger, collective break of roughly 10 min after half an hour, and smaller self-timed breaks between the remainder of the blocks.

[2]The source set is presented in three different subsets: train, validation, and test. We sampled from the validation and test subsets for MemCat, and from the train subset for the fillers images used in the memory task.

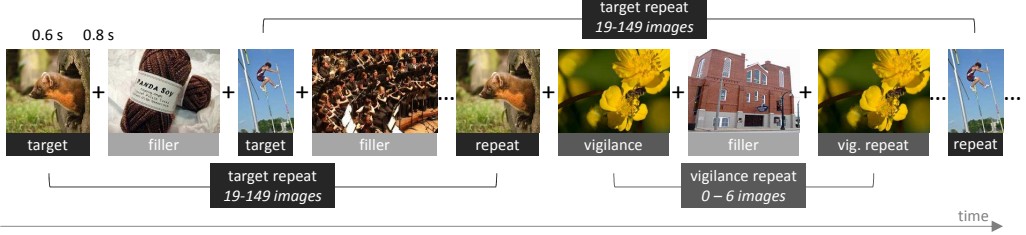

**Figure 3** Schematic of our implementation of the repeat-detection memory game first introduced by *Isola et al. (2014)*. Each image is presented for 600 ms, with an intertrial interval of 800 ms. Participants are instructed to press the space-bar whenever they recognize a repeat of a previously shown image.

Most students could complete eight blocks, but for some groups, slow data uploads at the end of a block resulted in lower numbers. AMT workers could complete one to 16 blocks, were allotted 48 h to submit their completed blocks (so, they were allowed to spread the blocks over time), and were paid $0.40 per block. To ensure a good quality of the AMT data and to avoid random or disingenuous responses, AMT workers were blocked from playing anymore blocks after two with a $d'$ lower than 1.5 on the vigilance trials. They were warned the first time this happened.

## Memorability measures

We computed two different, but related measures of memorability from the data collected through the repeat-detection memory game. These were the same two measures as used in LaMem, the largest available image memorability dataset yet. As mentioned in the Introduction, one measure is simply the proportion of participants recognizing the image when shown to them for the second time (i.e., the hit rate across participants). This is the "original" memorability measure, as introduced by *Isola et al. (2014)*, also adopted in many other memorability studies (e.g., *Bainbridge, Isola & Oliva, 2013*; *Bylinskii et al., 2015*; *Khosla et al., 2015*). The other memorability measure computed for the LaMem images was based on the same principle, but penalized for false alarms (i.e., when participants press the space-bar for the first presentation of the image) in the way proposed by *Khosla et al. (2013)*, who applied it to a dataset of face images (*Bainbridge, Isola & Oliva, 2013*). Rather than $H/N_{resp}$ (first measure), their formula was the following: $(H-F)/N_{resp}$, where H is the number of participants recognizing the image, F is the number of participants making a false alarm when the image is presented for the first time, and $N_{resp}$ is the total number of participants having been presented with the image. Here, an image's $N_{resp}$ was 99 (after exclusions) on average. Note that the memorability scores have an upper bound of 1 and a lower bound of 0. In theory, $(H-F)/N_{resp}$ could result in a negative score, but in practice it is highly unlikely that there would be more participants making a false alarm for the image than there are participants making a hit.

**Table 2  Recognition memory performance.** The table presents descriptive statistics across participants ($n = 2,291$) for five Signal Detection Theory measures. See *Macmillan & Creelman (2004)* for an explanation of these measures.

|        | $d'$ | $\beta$ | Hit rate | False alarm rate | Prop. correct |
|--------|------|---------|----------|------------------|---------------|
| Mean   | 2.50 | 4.43    | .76      | .05              | .87           |
| Median | 2.48 | 3.00    | .79      | .04              | .88           |
| SD     | 0.49 | 5.48    | .14      | .04              | .05           |
| Min    | 0.69 | 0.09    | .03      | .00              | .60           |
| Max    | 4.46 | 98.26   | 1.00     | .49              | .98           |

## RESULTS

### Participant performance

As mentioned, the performance on the easier vigilance trials was taken as an indication of whether participants were playing the memory game in a genuine way. If in a certain block, a participant did not distinguish vigilance repeats from non-repeat trials with a $d'$ of at least 1.5 (preset performance threshold), that block was excluded from further analyses. The exclusion rate amounted to 3% of all played blocks. Recall, however, that AMT workers were not allowed to play more blocks after two excluded ones.

After exclusion, the mean $d'$ across participants was 2.77 ($SD = 0.56$) for the vigilance repeats, and 2.47 ($SD = 0.50$) for the target repeats. Table 2 summarizes participants' overall performance, collapsing over vigilance and target repeats. Participants generally performed well on the task.

### Memorability scores

Participants' high performance was also reflected in the average image memorability scores. Figure 4 displays the mean for each of the two memorability measures as a horizontal line ($M_{H/Nresp} = .76$, $SD$; $M_{(H\text{-}F)/Nresp} = .70$). It is comparable to the mean observed in *Khosla et al. (2015)*. In addition, Fig. 4 visualizes the distribution of the collected image memorability scores for each of the five broad main categories separately. A simple linear regression revealed that the category explains 43% of the variance in the $H/N_{resp}$ scores and 44% of the variance in the $(H\text{-}F)/N_{resp}$. In line with previous research, the landscape images were on average the least memorable ($M_{H/Nresp} = .60$; $M_{(H\text{-}F)/Nresp} = .53$). They were followed by the vehicle images ($M_{H/Nresp} = .76$; $M_{(H\text{-}F)/Nresp} = .70$). Somewhat surprisingly, the food images generally came out on top of the ranking ($M_{H/Nresp} = .85$; $M_{(H\text{-}F)/Nresp} = .80$), topping the animal ($M_{H/Nresp} = .80$; $M_{(H\text{-}F)/Nresp} = .73$) and sports ($M_{H/Nresp} = .78$; $M_{(H\text{-}F)/Nresp} = .71$) categories. However, there is still a large degree of variability that is not explained by differences in broad image categories. Indeed, memorability varied considerably within categories as well, with $SDs$ of: .09 (animal; $SD_{(H\text{-}F)/Nresp} = .09$), .08 (food; $SD_{(H\text{-}F)/Nresp} = .08$), .13 (landscape; $SD_{(H\text{-}F)/Nresp} = .14$), .09 (sports; $SD_{(H\text{-}F)/Nresp} = .10$), and .09 (vehicle; $SD_{(H\text{-}F)/Nresp} = .09$).

Having observed that images from the same broader category indeed still differed in memorability, the next question was whether these differences are consistent across participants. This question taps into the reliability of the memorability measures. Following

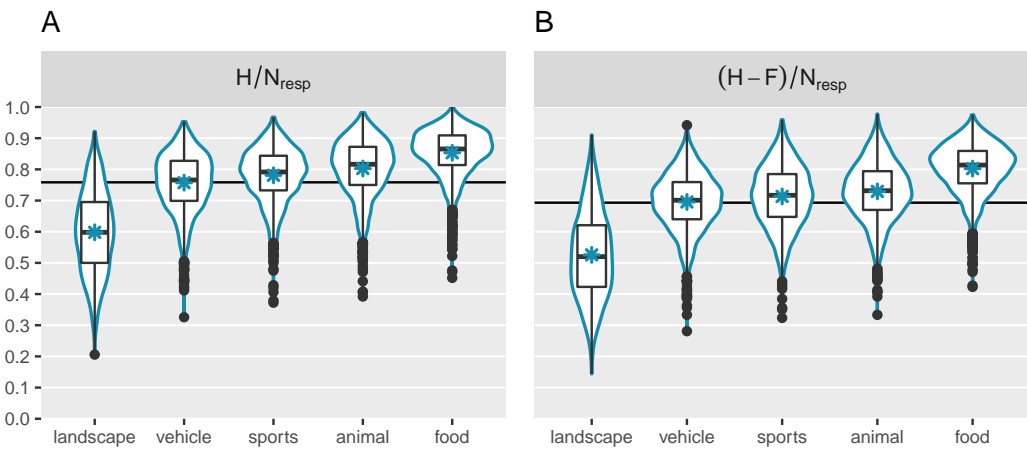

**Figure 4 Distribution of the collected memorability measures.** (A) Represents memorability scores computed as the hit rate across participants. (B) Represents scores corrected for false alarms. The horizontal lines indicate the global mean memorability scores. The asterisks represent the mean per category. Each category contains 2,000 quantified images. In addition to overall differences across categories, we observed considerable variability in memorability within categories too.

previous memorability work (e.g., *Isola et al., 2014*), the consistency was assessed by randomly splitting the participant pool in half, computing the memorability scores for each half separately and determining the Spearman's rank correlation between the two sets of scores. This was repeated for 1000 splits and the Spearman's rank correlation was averaged across the splits. Figure 5 shows the results in function of the mean $N_{resp}$ for each category as well as for the total image set.

We first discuss the results for $H/N_{resp}$ (see Fig. 5). When collapsing over all five categories, the observed mean split-half Spearman's rank correlation with all available responses ($N_{resp} = 99$, on average) amounted to .78. In comparison, *Khosla et al. (2015)* reported a mean split-half Spearman's rank correlation of .67 for their LaMem dataset. However, they only collected 80 responses per image. After randomly down-sampling our data to an $N_{resp}$ of 80, we still found a split-half consistency of .73. With the exception of the landscape category, for which we observed a total (i.e., without down-sampling) split-half consistency of .77, the total per category split-half consistency estimates were lower, ranging between .59 and .67. This is possibly due to smaller ranges of memorability scores within those categories (see Fig. 4). Note, however, that the split-half consistencies are an underestimate of the reliability of the memorability scores calculated based on the full participant pool. The latter can be estimated from the split-half consistency by means of the Spearman-Brown formula (*Brown, 1910*; *Spearman, 1910*). Applying this formula, we found the following final reliabilities for the $H/N_{resp}$ memorability scores: .87 (all), .80 (animal), .75 (food), .87 (landscape), .75 (sports), .78 (vehicle).

For the (H-F)/$N_{resp}$ memorability scores, we confine the discussion to pointing out that the pattern of results is qualitatively similar, although the final reliabilities are somewhat lower: .86 (all), .74 (animal), .71 (food), .85 (landscape), .77 (sports), .71 (vehicle).

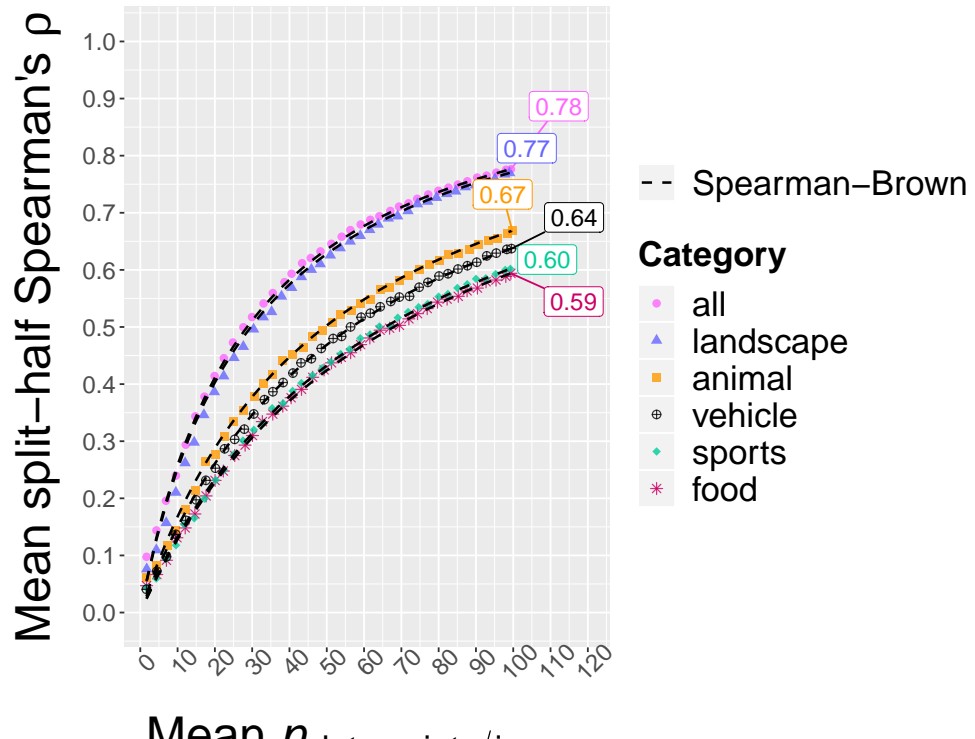

**Figure 5** **Split-half consistency across participants of the H/N$_{resp}$ memorability scores in function of N$_{resp}$.** Estimates are based on 1,000 random splits. N$_{resp}$ corresponds to the total number of data points for an image, not to the number that goes into one half during the split-half procedure. The dashed line represents predicted consistencies based on the Spearman–Brown formula (*Brown, 1910*; *Spearman, 1910*) applied to the observed consistency when N$_{resp}$ is the maximum number of available data points. Even though the consistency was lower when zooming in on a single category compared to the whole set at once (possibly due to a smaller range of scores), images still showed highly consistent differences in memorability within categories.

Finally, after finding that the two image memorability measures were both acceptably reliable, we asked how they compared to each other. In the current dataset, they were highly intercorrelated, as evidenced by a Pearson correlation of .93 when collapsing over all five categories. The per category correlations were: .82 (animal), .90 (food), .91 (landscape), .85 (sports), .88 (vehicle).

## DISCUSSION

We presented a new dataset, MemCat, consisting of a total of 10,000 images, each quantified on memorability using a repeat-detection memory task (first introduced by *Isola et al., 2014*, version used here based on *Khosla et al., 2015*). MemCat is the second largest image memorability dataset available, and the largest that is based on a category structure. That is, it is divided into five broader, memorability-relevant semantic categories: animal, food, landscape, sports, and vehicle, each with 2,000 exemplar images, which are further divided into subcategories (e.g., bear, cat, cow). Furthermore, the images were sampled

from popular, existing datasets such that additional annotations available there (e.g., segmentations masks or bounding boxes) would also be available to researchers wishing to use MemCat for research aimed at investigating specific factors underlying memorability.

Replicating previous research, we found that images differ considerably in memorability and that these differences are highly consistent across participants. Part but not all of this variability can be explained by differences between the five broader semantic categories. Note, however, that this result is correlational in nature, and that one should be cautious drawing causal conclusions. In line with *Bylinskii et al. (2015)*, considerable variability in memorability remained even within the categories. However, the consistency there was somewhat lower, probably because the variance was also lower. When the differences between images become smaller, it becomes harder to reliably and consistently distinguish them. Nevertheless, the consistency estimates per category were still high, indicating that we obtained reliable memorability scores. Finally, we reported results for two different methods to compute memorability scores. One is to compute the hit rate across participants: $H/N_{resp}$. This was the method used in the original work by *Isola et al. (2014)*. However, in principle, it possible that some images elicit more key presses not because they are truly recognized, but for some other reason (e.g., they seem familiar). That is why *Khosla et al. (2013)* suggested to correct for false alarms (i.e., when participants press the key for the first presentation of an image, when it is not a repeat) by computing $(H-F)/N_{resp}$. We report both measures for comparison, but note that they lead to a highly similar pattern of results and are also strongly intercorrelated. In what follows, we discuss possible uses of MemCat.

Most of what we learned from previous studies about what makes an image memorable is specified in terms of semantic categories or content types (e.g., images of people are more memorable than landscapes). However, a considerable amount of variability was still left unexplained. A primary use of the current dataset is in studies aiming to better understand the factors underlying image memorability. In particular, with 2,000 images for each of five broader categories, it allows to zoom in on variability within categories. This variability is of more interest to practical applications (e.g., advertising, education), because the semantic category or the content type (e.g., a certain product) will often be predefined and it will be a matter of choosing or creating a more memorable depiction of it. In addition to dividing the set into broad semantic categories, we also avoided variability due to other factors already discovered in previous studies (e.g., we excluded images depicting oddities, images containing text or recognizable places or faces), thus creating a set designed to help understand the previously unexplained variability in image memorability.

Second, MemCat is also useful as a benchmark for machine learning approaches to automatically predict memorability. Currently, LaMem (*Khosla et al., 2015*) is most often used, but models can now also be trained and tested on the current dataset. When taking *Khosla et al.*'s *(2015)* MemNet-CNN (without retraining), we found that its predictions show a rank correlation of .68 with the $(H-F)/N_{resp}$ memorability scores in the current set, suggesting that there is room for improvement. Given the category structure in MemCat, one could explore, for the first time, memorability models with one or more layers that are specific to a category. Indeed, it is possible that what makes landscape images memorable is different from what makes animal images memorable.

Finally, a third possible use is in neuroscientific studies or psychophysical studies examining effects of memorability. The current set offers a large number of quantified images to choose from. Moreover, it facilitates matching memorability conditions (e.g., high versus low) on semantic category, something that is often done in neuroscientific studies (e.g., *Bainbridge, Dilks & Oliva, 2017*; *Khaligh-Razavi et al., 2016*; *Mohsenzadeh et al., 2019*).

### Usage

On the MemCat project page (http://gestaltrevision.be/projects/memcat/), we provide a link to the collection of 10,000 images as well as links to two data files, all hosted on OSF (also see Additional Information). One file describes the images that were used and contains columns indicating the image filename in its source dataset, the name of its source dataset, the category (e.g., animal) and subcategory (e.g., bear) we assigned it to, the label that was used to sample it from its source dataset (e.g., American black bear), the current width, the current height, the factor by which it was resized (both the original width and height were multiplied by this factor), the number of hits (H), the number of false alarms (FA), the number of participants it was presented to ($N_{resp}$), and the two memorability scores. The other file contains the data collected in the repeat-detection memory game. Its columns indicate the participant ID (anonymized), the participant's age, the participant's gender, whether or not they participated through AMT, the block number, the trial number, the image shown, the trial type (target, target repeat, filler, vigilance, vigilance repeat), the participant's response (hit, correct rejection, miss, false alarm), the screen width, and the screen height.

## CONCLUSIONS

With MemCat, we present a large new dataset of 10,000 images fully annotated with ground truth memorability scores collected through an online repeat-detection memory task. It is the second largest memorability dataset to date and the largest with a hierarchical category structure. The results showed that images differ in memorability in ways that are consistent across participants, even within semantic categories. Among other things, MemCat allows the study of which factors might underlie such differences. Its richness in data and unique combination of features will appeal to a broad range of researchers in cognitive science and beyond (e.g., computer vision).

## ACKNOWLEDGEMENTS

The authors would like to thank the student-interns who have assisted in the image selection and data collection: Justine Aeyels and Joran Geeraerts, and Christophe Bossens, who is responsible for the technical support in the lab and contributed greatly to the implementation of the repeat-detection memory task on Amazon's Mechanical Turk.
## Funding

This work was supported by a personal fellowship by the Research Foundation –Flanders (FWO) awarded to Lore Goetschalckx (Grant 1108116N), and by a Methusalem grant awarded to Johan Wagemans by the Flemish Government (METH/14/02). The funders had no role in study design, data collection and analysis, decision to publish, or preparation of the manuscript.

## Grant Disclosures

The following grant information was disclosed by the authors:
Research Foundation—Flanders (FWO) awarded to Lore Goetschalckx: 1108116N.
Methusalem grant awarded to Johan Wagemans by the Flemish Government: METH/14/02.

## Competing Interests

The authors declare there are no competing interests.

## Author Contributions

- Lore Goetschalckx conceived and designed the experiments, performed the experiments, analyzed the data, contributed reagents/materials/analysis tools, prepared figures and/or tables, authored or reviewed drafts of the paper, approved the final draft.
- Johan Wagemans conceived and designed the experiments, authored or reviewed drafts of the paper, approved the final draft.

## Human Ethics

The following information was supplied relating to ethical approvals (i.e., approving body and any reference numbers):

The study was approved by SMEC, the Ethical Committee of the Division of Humanities and Social Sciences, KU Leuven, Belgium (Ethical Application Ref: G-2015 08 298).

## Data Availability

Goetschalckx, Lore, and Johan Wagemans. 2019. "MemCat: A New Category-Based Image Set Quantified on Memorability". OSF. June 17. https://osf.io/kvghu/.

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
