# Peer review of "MemCat: a new category-based image set quantified on memorability"

_PeerJ, doi:10.7717/peerj.8169_

## Round 0.1 · original submission · Minor Revisions

As you can see, both reviewers generally found the research to be good and the manuscript to be well written. Both had several minor comments, all of which I agree with. Please address each of these comments in your revision.

·

Basic reporting

Generally good. Please see the "small points" section of the general comments for a few suggestions.

Experimental design

Fine.

Validity of the findings

Please see items 2 and 3 of the general comments.

Additional comments

The authors provide a 10,000-image dataset that is organized by broad semantic category (5 superordinate-level categories) and normed in terms of memorability by an average of ~100 observers per image. This is a lovely effort and a great service to the community. I am generally pleased with the writeup of this work, but I do have a few suggestions.

-1- Motivating the database
As I stated, I believe that this database is a great tool for the field, but I feel that the uses and implications are not clearly stated early on in the introduction where most folks might start to read. For example, the framing of the manuscript in the first paragraph is a little unusual. Most of the cited papers are not studying how photographs “differ” (line 38) for their own sake, but studying how perception, attention, and memory work in the context of naturalistic visual inputs. I feel like the current motivation (to paraphrase: photography is popular, we should study it) actually cheapens the value of this work, which is of broad use to the cognitive science community. More broadly, the hierarchical category structure of the database is an untapped asset. The authors could ground this in terms of the cognitive science of hierarchical concepts (e.g. Rosch etc) and/or known neuroscientific concept distinctions such as animacy, real-world size, etc.

-2- Possible confound in sports category
Line 179-182: I am not following the logic here. I am assuming that the authors mean that non-incidental images of people were only allowed in the sports category. However, this might be an issue for two reasons: (1) if people are more intrinsically memorable, this could artificially inflate the memorability of this category; (2) if the people depicted in the sports images are famous, this would further artificially inflate memorability and/or increase variability in memorability because some (but not all) participants may recognize the players.

-3- Memorability scores
Lines 227-240: These memorability scores are somewhat non-standard. I am unsure of why the authors chose to combine two previous scores rather than more accepted precision/recall, and/or sensitivity/specificity.

-4- Black boxes
Lines 63-64: There are two types of “understanding” that the authors might be referring to, and it would provide clarity to know which one is intended. The “black box” argument often refers to the lack of human interpretability of dCNNs. However, dCNNs can also be critiqued for not having a full, semantic understanding of a picture. I suggest a clarification and a reference to help the reader understand which of these interpretations is intended.

-5- Small points
--Abstract, line 30 final sentence: suggested re-phrase “allows the study of” rather than “allows to study”
--Line 62: what is being predicted? I assume memorability score, but the sentence is a bit ambiguous
--Line 102: I suggest that the authors hyphenate “memory-relevant”
--Line 103: Citation needed about studies that show that these categories differ in terms of memorability.
--Line 112: No comma needed after “there”

·

Basic reporting

The narrative is easily understood and fits the scope of the work, all in all a well-written text with a clear structure. I only have three recommendations (ranked by importance from most to least important) for further improvement according to PeerJ guidelines.

1. Much of the current literature concerning memorability is presented although one important paper is ignored: https://ieeexplore.ieee.org/abstract/document/8704932. Ideally many members of the community will use this image set (it certainly deserves to be used). To increase understanding of memorability as a concept and how it may vary *within* semantic categories, I think it would be of benefit if the potential role of Visual Memory Schema is shortly addressed in the introduction, for instance in page 4 after line 80 when goodness of image organisation is introduced.
2. The concept of memorability relevance (page 5, line 96) is introduced but only explained three sentences later. Comprehensibility of the paragraph would be improved if memorability relevance were explained immediately.
3. Data visualisations are informative and aesthetically pleasing. Captions only include basic descriptions of what is shown but results could be communicated even more effectively if captions include a short summary of the meaning of the figure/results.

Experimental design

The methods are very well explained and fit the overarching research question. No further comment.

Validity of the findings

Analyses are statistically appropriate and accessibility to and composition of the stimulus set and (raw) data files are exemplary. No further comment.

---

## Round 0.2 · accepted · Accept

As you can see, both authors felt that your changes sufficiently addressed their concerns, and I agree with them.

I would note a few small changes for reading clarity that I would recommend that you correct at the proofing stage (optional, of course):
ln 38: “make abstraction of” change to “abstract away from the properties of”
ln 118: delete “about it”
ln 127: change “made them” to “helped them” ?

·

Basic reporting

No comment.

Experimental design

No comment.

Validity of the findings

No comment.

Additional comments

I would like to thank the authors for their revision. I feel that the new structure of the paper will give it the attention it deserves.

---

## Author Rebuttal · Round 0.2

Dear Editor,

Dear Reviewers,

Please find enclosed our revised manuscript titled: "MemCat: A new category-based image set quantified on memorability" by Lore Goetschalckx and Johan Wagemans, which we would like to resubmit for publication in *PeerJ.* We are grateful to the reviewers for their constructive comments and provide a response to each of their points below. We are convinced that our manuscript has improved significantly compared to the first submission, thanks to the reviewers' feedback, and we hope that you will find that is now suitable for publication *PeerJ*.

Sincerely,

Lore Goetschalckx and Johan Wagemans
* * *
**Reviewer 1 – Michelle Greene**

1. *Motivating the database. As I stated, I believe that this database is a great tool for the field, but I feel that the uses and implications are not clearly stated early on in the introduction where most folks might start to read. For example, the framing of the manuscript in the first paragraph is a little unusual. Most of the cited papers are not studying how photographs "differ" (line 38) for their own sake, but studying how perception, attention, and memory work in the context of naturalistic visual inputs. I feel like the current motivation (to paraphrase: photography is popular, we should study it) actually cheapens the value of this work, which is of broad use to the cognitive science community. More broadly, the hierarchical category structure of the database is an untapped asset. The authors could ground this in terms of the cognitive science of hierarchical concepts (e.g. Rosch etc) and/or known neuroscientific concept distinctions such as animacy, real-world size, etc.*

   → We thank the reviewer for acknowledging the strengths of our work and for pointing out how to emphasize them better in an improved introduction. Following the reviewer's advice, we have replaced the first paragraph of the Introduction to avoid the motivation coming across as just "photography is popular, we should study it". Furthermore, we added a new paragraph attesting to the central position of (hierarchical) categories in the broader cognitive system (citing, among others, Rosch and neuroscientific studies, as recommended), thereby providing a more solid ground for the added value of the hierarchical category structure that characterizes MemCat. The paragraph starts around Line 80). Finally, we now offer a preview of the implications and uses of MemCat in the last paragraph of the Introduction (still referring to the Discussion for more elaboration) and emphasize once more that MemCat's hierarchical category structure is of broad use to the cognitive science community.

[Figure]

[Figure]

2. *Possible confound in sports category. Line 179-182: I am not following the logic here. I am assuming that the authors mean that non-incidental images of people were only allowed in the sports category. However, this might be an issue for two reasons: (1) if people are more intrinsically memorable, this could artificially inflate the memorability of this category; (2) if the people depicted in the sports images are famous, this would further artificially inflate memorability and/or increase variability in memorability because some (but not all) participants may recognize the players.*

   → Reply to point (1): We agree that previous research has suggested that images depicting people tend to be memorable. However, that was exactly the reason we wanted to include at least one people category. It is "memorability-relevant", as we have put it. In the same vein, we included, for example, a separate landscape category, because previous findings suggested that they are typically less memorable overall. Because we deemed "people" in general too broad of a category compared to the others and also because it did not lend itself well to an intuitive division into subcategories, we chose to narrow it down to "sports" (i.e., images of people practicing sports). In other words, "sports" was our designated people category. The presence of people is not a confound, as it is not so much that we *allowed* the non-incidental presence of people but  this was instead the number one inclusion criterion for this category. So, all the sports images have one or more people in it practicing sports (examples shown in Figure 1, full image set available through gestaltrevision.be/projects/memcat). An image that does not satisfy this criterion is in fact not a sports image in our logic, although we acknowledge that perhaps one can consider an image of an empty basketball court a sports image too (but we did not include such images). To clarify our reasoning/motivation behind the sports category, we have elaborated on this in the Introduction (roughly around Line 135) as well as in the Materials section (roughly around Line 228).

   → Reply to point (2): We fully agree that if an image depicts a famous sports player, this could inflate its memorability score. It is for this reason that we tried to avoid such images to the best of our ability, as mentioned around Line 222: "Similarly, we avoided images depicting famous places or people…" During the manual selection phase, we specifically paid attention to this and avoided images of, for example, Roger Federer (tennis) and Cristiano Ronaldo (soccer). These examples are now added to the manuscript (near Line 223). Of course, this is certainly not 100% error proof, as we ourselves probably do not know all famous sports players. For the COCO images, we tried to rely also on the image captions available from the original dataset (i.e. Amazon's Mechanical Turk workers describing what they see in the image). If none of the captions mentioned the player's name, we felt supported in our decision that the player was not famous. Based on these precautions as well as the fact that definitely not all of the images depicted professional looking players (as indicated by their gear, large crowds watching, big gyms, etc.), we feel that the possible confound was reduced to a minimum.

[Figure] [Figure]

3. *Memorability scores. Lines 227-240: These memorability scores are somewhat non-standard. I am unsure of why the authors chose to combine two previous scores rather than more accepted precision/recall, and/or sensitivity/specificity.*

→ To clarify, we did not introduce a new way of computing memorability scores. Both the measures we used, $H/N_{resp}$ and $(H-F)/N_{resp}$, have been used in previous memorability work. Moreover, they are the same two measures that were also computed for the images in LaMem (Khosla et al., 2015), the largest memorability dataset available yet.

→ The first measure, $H/N_{resp}$, is the "original" memorability measure, as introduced by Isola et al. (2014), also adopted in many other memorability studies (e.g., Bainbridge, Isola, & Oliva, 2013; Bylinskii, Isola, Bainbridge, Torralba, & Oliva, 2015; Khosla, Raju, Torralba, & Oliva, 2015). It is the proportion of participants recognizing the image upon its repeat (H) out of all participants who were shown this image ($N_{resp}$). It is the hit rate (**recall, sensitivity**) per image, across participants.

→ Note that MemCat images always functioned as targets (never fillers) and if a sequence of the memory game contained a certain MemCat image, it also contained its repeat. So, the number of times a MemCat image should not elicit a key press (because it is the first presentation; negative cases) and the number of times a MemCat image should elicit a key press (because it is a repeat and participants should recognize it; positive cases) are equal. We refer to this number as $N_{resp}$. On Line 283 the definition of $N_{resp}$ is phrased as: "$N_{resp}$ is the total number of participants having been presented with the image".

→ The second measure, $(H-F)/N_{resp}$ penalizes for those cases where an image elicits a key press where it should not, because it is only the first presentation. This would be a false alarm (or a false positive). This measure was also used in Khosla et al. (2013) and Khosla et al. (2015). Note that $F/N_{resp}$ would be the false alarm rate (1-**specificity**) per image, across participants.

→ In sum, we are not proposing new memorability measures. Both of the measures we report follow previous memorability work. We have adjusted the Memorability Measures section (Methods) to make this point more clear and to avoid confusion. Furthermore, both measures are very much related to precision/recall, and/or sensitivity/specificity. Finally, the full, raw data collected through the memory game are available through http://gestaltrevision.be/projects/memcat/, allowing researchers to compute alternative memorability scores, if desired.

4.  Lines 63-64: There are two types of "understanding" that the authors might be referring to, and it would provide clarity to know which one is intended. The "black box" argument often refers to the lack of human interpretability of dCNNs. However, dCNNs can also be critiqued for not having a full, semantic understanding of a picture. I suggest a clarification and a reference to help the reader understand which of these *interpretations* is intended.

    → We thank the reviewer for pointing out the ambiguity. We were mostly referring to the first type and have now clarified that in the text, as well as added a reference. The second is also a valid critique of CNNs but less relevant to the present discussion.

5.  Small points.

    - *Abstract, line 30 final sentence: suggested re-phrase "allows the study of" rather than "allows to study"* → Thanks, we have rephrased it.
    - *Line 62: what is being predicted? I assume memorability score, but the sentence is a bit ambiguous* → Thanks, we have specified it further in the text.
    - *Line 102: I suggest that the authors hyphenate "memory-relevant"* → Thanks, we have added the hyphens.
    - *Line 103: Citation needed about studies that show that these categories differ in terms of memorability.* → Agreed, we have added the citation.
    - *Line 112: No comma needed after "there"* → Thanks, we have removed the comma.

**Reviewer 1 – Nico Broers**

1.  *Much of the current literature concerning memorability is presented although one important paper is ignored: https://ieeexplore.ieee.org/abstract/document/8704932. Ideally many members of the community will use this image set (it certainly deserves to be used). To increase understanding of memorability as a concept and how it may vary \*within\* semantic categories, I think it would be of benefit if the potential role of Visual Memory Schema is shortly addressed in the introduction, for instance in page 4 after line 80 when goodness of image organisation is introduced.*

    → We thank the reviewer for bringing this valuable memorability study to our attention. We have now cited it in the Introduction (Line 90). We felt that it fitted best in the paragraph about the central position of categories in the broader cognitive system (i.e., the new paragraph added to the Introduction following Reviewer 1's advice).

2.  *The concept of memorability relevance (page 5, line 96) is introduced but only explained three sentences later. Comprehensibility of the paragraph would be improved if memorability relevance were explained immediately.*

    → We have adjusted the text such that the notion of "memorability relevance" is only introduced where it is explained.

[Figure]
[Figure]

3. *Data visualisations are informative and aesthetically pleasing. Captions only include basic descriptions of what is shown but results could be communicated even more effectively if captions include a short summary of the meaning of the figure/results.*

→ We thank the review for pointing out how to improve the captions and have now added summaries where appropriate.